# Ashwagandha Ethanol Extract Attenuates Sarcopenia-Related Muscle Atrophy in Aged Mice

**DOI:** 10.3390/nu16010157

**Published:** 2024-01-03

**Authors:** Jin-Sung Ko, Bo-Yoon Chang, Young-Ju Choi, Ji-Soo Choi, Hee-Yeon Kwon, Jae-Yeon Lee, Sung-Yeon Kim, Se-Young Choung

**Affiliations:** 1Department of Biomedical Science & BK21 Four NBM Global Research Center for Regenerative Medicine, Dankook University, Cheonan 31116, Chungnam, Republic of Korea; 72170428@dankook.ac.kr (J.-S.K.); breadjj00@dankook.ac.kr (Y.-J.C.); 2Institute of Pharmaceutical Research and Development, College of Pharmacy, Wonkwang University, Iksan 54538, Jeonbuk, Republic of Korea; pama611@naver.com (B.-Y.C.); sungykim@wku.ac.kr (S.-Y.K.); 3R&D Center, NSTbio Co., Ltd., 32 Songdogwahak-ro, Yeonsu-gu, Incheon 21984, Republic of Korea; cjs116@nstbio.co.kr (J.-S.C.); khy@nstbio.co.kr (H.-Y.K.); jyleebio@nstbio.co.kr (J.-Y.L.); 4Department of Preventive Pharmacy, College of Pharmacy, Dankook University, Cheonan 31116, Chungnam, Republic of Korea

**Keywords:** *Withania somnifera* extract, sarcopenia, muscle atrophy, in vivo, in vitro

## Abstract

The investigation focused on the impact of *Withania somnifera* (ashwagandha) extract (WSE) on age-related mechanisms affecting skeletal muscle sarcopenia-related muscle atrophy in aged mice. Beyond evaluating muscular aspects, the study explored chronic low-grade inflammation, muscle regeneration, and mitochondrial biogenesis. WSE administration, in comparison to the control group, demonstrated no significant differences in body weight, diet, or water intake, affirming its safety profile. Notably, WSE exhibited a propensity to reduce epidermal and abdominal fat while significantly increasing muscle mass at a dosage of 200 mg/kg. The muscle-to-fat ratio, adjusted for body weight, increased across all treatment groups. WSE administration led to a reduction in the pro-inflammatory cytokines TNF-α and IL-1β, mitigating inflammation-associated muscle atrophy. In a 12-month-old mouse model equivalent to a 50-year-old human, WSE effectively preserved muscle strength, stabilized grip strength, and increased muscle tissue weight. Positive effects were observed in running performance and endurance. Mechanistically, WSE balanced muscle protein synthesis/degradation, promoted fiber differentiation, and enhanced mitochondrial biogenesis through the IGF-1/Akt/mTOR pathway. This study provides compelling evidence for the anti-sarcopenic effects of WSE, positioning it as a promising candidate for preventing sarcopenia pending further clinical validation.

## 1. Introduction

Sarcopenia, much like other complex syndromes, arises from a myriad of interrelated mechanisms. These mechanisms are associated with aging, inadequate nutrition, physical inactivity, and endocrine imbalances [1,2]. During the aging process, a decline in essential hormones such as testosterone, insulin-like growth factor (IGF-1), and estrogen, which play a crucial role in regulating muscle protein synthesis, becomes evident [3,4].

Moreover, sarcopenia is linked to alterations in skeletal muscle physiology and cellular mechanisms. These changes encompass metabolic, cellular, vascular, and inflammatory levels. Notably, higher levels of inflammatory markers have been identified as detrimental to skeletal muscle metabolism, whether through direct catabolic effects or reductions in growth hormone [5,6]. Myofibrillar protein synthesis is hindered due to satellite cells failing to respond positively to growth factors and cytokines (myokines), which are essential for stimulating the production of contractile proteins [7].

Research has demonstrated that in the elderly, the concentration of inflammatory markers, including CRP, IL-6, and TNF-α, increases while physical performance concurrently decreases [8,9]. Oxidative stress and inflammation are characteristic features of age-related muscle atrophy, presenting potential targets for therapeutic interventions. Recent studies underscore the pivotal role of inflammation in maintaining skeletal muscle homeostasis and influencing the mechanisms that lead to sarcopenia [1,10]. Acute inflammation models are being employed to explore the molecular pathways linking inflammation to muscle protein metabolism [11,12]. These studies have revealed that inflammatory signaling triggers muscle catabolism, and reducing inflammation has shown promise for enhancing muscle performance.

However, beyond pathological conditions, low-grade inflammation is associated with the aging process [13]. Assessing muscle performance within the realm of natural aging is crucial when considering candidates for the prevention of age-related sarcopenia, as opposed to assessing it in the context of acute, high levels of inflammation [14].

*Withania somnifera* (L.) Dunal, commonly known as ashwagandha, belongs to the Solanaceae family and has been a botanical remedy with a history spanning centuries [15]. It has been employed in the treatment of various chronic conditions, including hypertension [16], arthritis [17], diabetes [18], Alzheimer’s disease [19], and depression [20]. In the realm of holistic health, it is often likened to ginseng for its stress-reducing capabilities, cognitive enhancements (e.g., memory improvement), and support for a robust immune system [21]. One of its remarkable features is its ability to modulate the immune system, striking a balance between the humoral and cellular responses of the adaptive immune system [22]. The therapeutic effects of *W. somnifera* are primarily attributed to its active constituents, including alkaloids, steroidal lactones (e.g., withanolides, withaferin A), and steroidal saponins [23,24]. Despite its well-established safety and recommendations for treating various ailments, our current understanding of its role in ameliorating senile sarcopenia and the underlying mechanisms remains limited.

In this study, our objective was to assess the efficacy of the bioactive constituents derived from *W. somnifera* in mitigating dexamethasone (DEX)-induced muscle atrophy in C2C12 myotubes. Furthermore, we aimed to elucidate the molecular mechanisms underlying the effects of *W. somnifera* on sarcopenia induced by low-grade inflammation, a condition associated with natural aging in C57BL/6 mice.

## 2. Materials and Methods

### 2.1. Preparation of WSE

Roots and leaves of *Withania somnifera* (ashwagandha) were purchased from PICASSO GLOBAL LLP (Mumbai, Maharashtra, India). To produce the mixed extract of ashwagandha, 934 g of roots and 66 g of leaves were extracted with 10 L of 40% fermented ethanol, followed by soaking for over 15 h. After this, the extraction process was performed for 4 h at 50 °C. The extracted solution was filtered and concentrated to a brix level of 15 or higher, and then maltodextrin was added in an amount equal to the solid content of the concentrate. After sterilization at 65 °C for 30 min, it was spray-dried. The dried powder was sealed for use in experiments. This mixed extract of ashwagandha was named WSE.

### 2.2. The Analysis of Withanolide A Using HPLC

The identification and quantification of withanolide A in WSE were analyzed using an HPLC system (Shimadzu Corporation, Kyoto, Japan) equipped with an LC-20AD series pumping system and an SPD-M20A photodiode array detector (PDA). Separation was carried out on a symmetry column (250 × 4.6 mm, 5 μm), and the column temperature was maintained at 20 °C. The binary mobile phase consisted of water (solvent A) and acetonitrile (solvent B). The flow rate was kept constant at 1.0 mL/min for a total run time of 60 min. The mobile phase was run with a gradient program: 0–5 min, 20% B; 5–20 min, 50% B; 20–30 min, 50% B; 30–50 min, 100% B; 50–60 min, 100% B. The sample injection volume was 20 µL. The peak of interest was monitored at 190–380 nm by a PDA detector, and the spectra were compared with the standard.

### 2.3. Mice and Design of the Animal Experiment

Forty-eight-week-old male C57BL/6J mice were housed in the SPF animal room of Dankook University and maintained at a constant temperature (23 ± 2 °C) with a 12-h light/dark cycle. Following a two-week acclimatization period, animals without any anomalies in weight gain or general behavior were chosen and allocated into groups, ensuring uniform average body weight distribution among them. The mice were categorized into groups of 10 mice each, and the groups were administered as follows: 1. Vehicle (control; CON); 2. *S. chinensis* extract (SCE) 200 mg/kg; 3. WSE 100 mg/kg; 4. WSE 200 mg/kg; 5. WSE 300 mg/kg. The candidate was orally administered at a dosage of 10 mL/kg once daily for a duration of 8 weeks. Body weight and grip strength were assessed at 3-day intervals throughout the study, and an exhaustion test was conducted one day prior to the study end point. Following the sacrifice, we procured blood samples, leg muscle tissues, including quadriceps, gastrocnemius, and soleus, and fat tissues. The animal study was conducted in compliance with the guidelines set forth by the Institutional Animal Care and Use Committee of Dankook University, with approval number DKU-23-027.

### 2.4. Assessment of Serum ALT and AST

To confirm the safety of WSE administration, we investigated Alanine Aminotransferase (ALT) and Aspartate Aminotransferase (AST) levels in serum. At the end of the experiment, the serum was obtained from the inferior vena cava of mice. ALT and AST levels were quantified colorimetrically using enzymatic kits (Asan Co., Seoul, Republic of Korea).

### 2.5. Measurement of Grip Strength

The experiment involved assessing the grip strength of mice’s front paws using a grip strength meter (Jeung Do Bio & Plant, Seoul, Republic of Korea) at three-day intervals. The mice were positioned on a grid connected to the grip strength meter, and their maximum grip force was recorded by pulling their tails while they held onto the grid. To ensure accuracy, each test was repeated three times, and the resulting averages were normalized by the mice’s body weight for subsequent analysis.

### 2.6. Measurement of Exhaustion on the Treadmill

The mice underwent a 3-day acclimatization process on the treadmill (Jeung Do Bio & Plant, Seoul, Republic of Korea). This acclimatization involved starting at a speed of 10 m/min and increasing it by 3 m/min at 10-min intervals. Subsequently, the mice were assessed for exhaustion. In the exhaustion test, the treadmill was initiated at a speed of 12 m/min and subsequently raised by 3 m/min every 3 min until it reached a final speed of 30 m/min. The exhaustion point on the treadmill was defined as the moment when a mouse could not sustain the pace for over 10 s.

### 2.7. Assessment of Serum Cytokine Levels

We used serum obtained from the inferior vena cava of the mice. The concentrations of TNF-α, IL-1, and IL-6 in the serum were determined using an immunoassay kit from R&D Systems (Minneapolis, MN, USA). The assessments were carried out following the protocols provided by the respective suppliers and measured with an ELISA microplate reader from BioTek Instruments Inc. (Vermont, WI, USA).

### 2.8. Histological Analysis

The gastrocnemius tissues were initially fixed in a 4% formalin solution and subsequently subjected to dehydration in a 30% sucrose solution. Following this, the tissues underwent cryo-embedding utilizing a cryogel (Leica, Microsystems Europe, Breckland, UK). Cryo-tissues were each sectioned at 5 μm in thickness and subsequently stained using hematoxylin and eosin. The calculation of the cross-sectional area (CSA) was performed under a light microscope at 200× magnification, and the dimensions of each image were measured using the Image J software program (Institutes of Health, Bethesda, MA, USA, https://imagej.nih.gov/ij/download.html, accessed on 27 April 2023).

### 2.9. Cell Culture and Differentiation

C2C12 myoblasts obtained from ATCC (USA) were cultured in Dulbecco’s Modified Eagle Medium (DMEM; ATCC, Manassas, VA, USA) supplemented with 10% fetal bovine serum (FBS; Gibco, Grand Island, NY, USA) and 1% penicillin-streptomycin (PS; Welgene, Gyeongsan-si, Republic of Korea). To induce differentiation, cells were allowed to grow until they reached 70 to 80% confluence, at which point the medium was switched to DMEM supplemented with horse serum (HS; Gibco, USA) and 1% PS. Cells were then cultured for 7 days to promote differentiation. The culture was maintained in a humidified incubator with 5% CO_2_ at 37 °C, and the culture medium was refreshed every 2 days to ensure optimal conditions for the cells.

### 2.10. Measurement of Myotube Diameter

C2C12 cells were cultured in 6-well plates and induced to differentiate into myotubes by incubation in 2% horse serum-containing medium for 7 days once they reached 80% confluency. Subsequently, they were exposed to a 48-h treatment involving a combination of ashwagandha extract and 50 μM dexamethasone. After this treatment, the cells were thoroughly washed with PBS and then fixed with 100% methanol for 5 min, followed by a 10-min air-drying step to remove the methanol. Next, Jenner staining solution, diluted threefold with 1 mM sodium phosphate buffer (pH 5.6), was applied for 5 min and washed twice with distilled water. This was followed by the use of Giemsa staining solution, diluted twenty-fold with 1 mM sodium phosphate buffer (pH 5.6), for 10 min at room temperature, with a subsequent wash using distilled water. The thickness of the myotubes was imaged at 200× magnification using a light microscope equipped with a CCD camera (CKX53, Olympus, Tokyo, Japan) and analyzed with Image J software (USA).

### 2.11. Protein Expression Analysis

The cell or gastrocnemius muscle tissue was homogenized in a lysis buffer containing cOmplete™ Protease Inhibitor Cocktail (Roche Diagnostics, Indianapolis, IN, USA) and protease inhibitor (Sigma-Aldrich, St. Louis, MO, USA; Oakville, ON, Canada) and subsequently centrifuged. The BCA Protein Assay Kit (Bio-Rad, Hercules, CA, USA) was employed to determine the protein concentrations, ensuring an equal protein concentration across all experimental groups. The equivalent protein concentrations from each group were then subjected to SDS-PAGE. Following electrophoresis, the membranes were blocked with 5% skim milk and incubated with primary antibodies overnight. The primary antibodies utilized for Western blot analysis included IGF, *p*-AKT, t-AKT, *p*-mTOR, mTOR, *p*-PI3K, t-PI3K, MyoD, Myogenin, MuRF1, *p*-FOXO3a, FOXO3a, Sirt1, and PGC1α (Santa Cruz Biotechnology, Santa Cruz, CA, USA), as well as GAPDH and β-actin (Sigma-Aldrich, CA, USA) for use as a loading control. Subsequently, the membranes were incubated with the corresponding secondary antibodies to visualize the protein bands using an LAS3000 luminescent image analyzer (Fuji Film, Tokyo, Japan). β-actin served as the loading control, and Image J software (National Institute of Health, Bethesda, MD, USA) was employed for quantitative analysis.

### 2.12. Gene Expression Analysis

Total RNA was isolated from the cell or gastrocnemius muscle using the easy-Blue™ kit (iNtRON, Seongnam, Republic of Korea), followed by cDNA synthesis with the High Capacity RNA-to-cDNA kit (Thermo Fisher Scientific, Waltham, MA, USA). Gene expression levels were quantified by combining template DNA with specific primers (Myogenin, MyoD, MuRF1, Atrogin-1, Myostatin, SIRT1, and PGC-1α) using the AccuPrep^®^ Genomic DNA Extraction Kit (Bioneer, Daejeon, Republic of Korea). The quantification was performed through 40 cycles, consisting of incubation at 48 °C for 15 min, 95 °C for 10 min, 95 °C for 15 s, and 60 °C for 1 min. Data analysis was performed using StepOne Software v2.3. To normalize the actin housekeeping gene expression, the mathematical model of the relative expression ratio, including PCR efficiency, was chosen and applied for sample quantification (Table 1).

### 2.13. Statistical Analysis

Data are expressed as the mean ± SE. Significant differences were compared using a one-way ANOVA, followed by a Tukey’s test. Statistical significance was defined as *p* < 0.05. All statistical analyses were performed using GraphPad Prism v.5.0 (Chicago, IL, USA).

## 3. Results

### 3.1. Standardization of WSE

To identify and quantify withanolide A in WSE, the component was analyzed using HPLC and compared to a standard material of withanolde A (PhytoLab, Vestenbergsgreuth, Germany). The chromatogram of WSE was identified by comparing the retention time (RT) with that of the standard withanolide A at 225 nm. The HPLC results showed that withanolide A was present in WSE with a retention time of 26.08, and this was compared to the standard material with a retention time of 22.03. Additionally, the spectra of WSE and the standard material of withanolide A were found to be highly similar. Using the standard curve for comparison, the amount of withanolide A in WSE was determined to be 0.55 mg/g (Figure 1).

### 3.2. Effect of WSE on Muscle Performance in Aged Mice

At the onset of the study, 50-week-old C57BL6/J mice exhibited an initial weight of 34.2 ± 2.1 g. The daily food intake was observed to be 2.78 ± 0.2 g/mouse, with no statistically significant differences in food intake or body weight detected between the groups subjected to SCE and WSE treatments by the end of the study (Figure 2A,B). The assessment of hepatotoxicity induced by the administration of the test substance involved the measurement of serum ALT and AST levels. In the CON group, the ALT was 49.8 ± 2.80 Unit/mL, and the AST was 50.7 ± 5.24 Unit/mL. Following the administration of the SCE and WSE, no significant differences in ALT and AST levels were observed when compared to the CON group. These findings suggest that the tested substances, SCE and WSE, did not show a significant change in serum ALT and AST levels, indicating no hepatotoxic effects under the conditions of the study (Figure 2C). The grip strength of the mice’s forelimbs exhibited a declining trend in all groups during the initial second week of the study. The results showed a significant difference between CON and the other groups at week 6. At week 8, SCE (3.1 ± 0.1 g/g) displayed a substantial 29.3% increase in grip strength compared to the CON (2.4 ± 0.1 g/g) group. WSE at doses of 100 (2.9 ± 0.1 g/g), 200 (3.2 ± 0.1 g/g), and 300 (3.0 ± 0.2 g/kg) mg/kg exhibited grip strength enhancements of 19.6%, 32.8%, and 23.3% relative to the CON group, respectively (Figure 2D). The running time to exhaustion for the CON group was 19.0 ± 3.4 min, covering a distance of 392.2 ± 100.5 m. In comparison, the running time and distance for the WSE 100 mg/kg group were 21.4 ± 3.4 min and 459.6 ± 100.6 m, respectively, and for the 200 mg/kg group, they were 22.7 ± 4.9 min and 506.4 ± 142.4 m, respectively. Although there was an observable trend towards increased running time and distance in the WSE treated at 200 mg/kg when compared to the CON, these differences were not statistically significant (Figure 2E,F).

### 3.3. Effect of WSE on Chronic Low-Grade Inflammation in Aged Mice

To assess the impact of WSE administration on serum inflammation in aged mice, the expression of cytokines was evaluated. While no significant differences were noted in the levels of IL-6 across the treatment groups, both TNF-α and IL-1β exhibited significant reductions in mice treated with WSE and SCE compared to the CON (Figure 3A). The mRNA levels of cytokines in the gastrocnemius muscle showed a significant decrease in TNF-α at 200 and 300 mg/kg of WSE compared to the CON, and a significant decrease in IL-1β and IL-6 at all concentrations of WSE compared to the CON (Figure 3B).

### 3.4. Effect of WSE on Muscle Mass and Myofiber Cross-Sectional Area (CSA) in Aged Mice

The muscle weights of fast-twitch fibers, including quadriceps and gastrocnemius, as well as slow-twitch fiber muscles like the soleus, exhibited a tendency to increase in the WSE-treated group compared to the control group. However, it is noteworthy that only the gastrocnemius muscle in the WSE 200 mg/kg group displayed a statistically significant increase of 6.03% compared to the CON (0.53 ± 0.01%). In the CON group, the relative weights of epididymal and abdominal fat were 1.88 ± 0.11% and 0.76 ± 0.05%, respectively. Notably, there was a consistent decrease in the relative weights of both types of fat (epididymal and abdominal) in the groups treated with SCE and WSE. Specifically, the WSE 200 mg/kg treatment group exhibited significant reductions, with relative weights of 1.45 ± 0.46% for epididymal fat and 0.54 ± 0.19% for abdominal fat, when compared to the CON group (Figure 4). The cross-sectional area (CSA) of the gastrocnemius muscle exhibited a similar trend as muscle mass, showing an increase. In the CSA distribution graph, the control group’s muscle fibers were predominantly clustered within the range of 500 to 1500 µM^2^. In contrast, the groups administered WSE at a dosage of 200 mg/kg displayed a substantial increase in muscle fibers, with the CSA exceeding 1500 µM^2^, resulting in a broader distribution of the CSA.

### 3.5. Effect of WSE on Muscle Protein Synthesis and Proteolysis through the AKT/mTOR Pathway in Aged Mice

MyoD, characterized as a primary myogenic regulator, exhibited an increase in protein expression following treatment with both WSE and SCE, with a notable upregulation observed with WSE at 200 mg/kg compared to the CON. As for Myogenin, a secondary myogenic regulator, treatment with WSE at doses of 100 mg/kg, 200 mg/kg, and 300 mg/kg yielded significant increases in protein expression compared to the CON (Figure 5C,D). Conversely, with regard to muscle protein proteolytic factors, specifically *p*-FOXO3a, MuRF1, Myostatin, and Atrogin-1, no significant reduction was evident following treatment with WSE and SCE (Figure 5E,F). In WSE treatment, a discernible elevation in IGF-1 protein levels was observed when compared to the CON, signifying AKT phosphorylation. The initial validation is the activation of AKT and mTOR, pivotal components situated upstream in the signaling pathway responsible for muscle protein synthesis in aged mice (Figure 5A,B).

### 3.6. Effect of WSE on Mitochondrial Biogenesis through the SIRT1/PGC-1α Pathway in Aged Mice

Sirt1, a pivotal factor implicated in the promotion of mitochondrial biosynthesis, exhibited noteworthy increases in both protein and mRNA expression levels under the influence of WSE, particularly in relation to PGC1α (Figure 6).

### 3.7. Effect of WSE on the Formation of Myotubes in C2C12

C2C12 cells were induced to differentiate into myotubes and then treated with varying concentrations (12.5, 25, 50, 100, and 200 μg/mL) of WSE and SCE. The cytotoxicity assay revealed a proliferative effect at 50 μg/mL of WSE, while cytotoxicity was observed in the 200 μg/mL treatment group. SCE exhibited no cytotoxicity at any concentration. Subsequent experiments were performed with 25, 50, and 100 μg/mL of WSE and 100 μg/mL of the positive control SCE. The following experiments were performed with 25, 50, and 100 μg/mL of WSE and 100 μg/mL of the positive control SCE (Figure 7A). After inducing muscle atrophy through treatment with dexamethasone (DEX) at 50 μM, the analysis of myotube diameter was conducted to assess the potential of WSE in ameliorating muscle atrophy. The DEX group (19.26 μm) showed a 33.92% reduction in root canal thickness compared to the CON group (29.14 μm). Under the conditions of induced muscle atrophy, WSE at 25, 50, and 100 μg/mL demonstrated a concentration-dependent increase in myotube diameter, measuring 22.46, 22.78, and 25.79 μm, respectively. Notably, these values showed a significant increase compared to the DEX group. In contrast, *S. chinensis*, as a positive control, did not exhibit a significant increase compared to the DEX group (Figure 7B).

### 3.8. Effect of WSE on Muscle Protein Synthesis and Protein Degradation through the PI3K/Akt Pathway in Dexamethasone-Induced C2C12 Muscle Atrophy

In the DEX-induced muscle atrophy group, there was a significant 0.52-fold decrease in the *p*-mTOR/t-mTOR ratio, a 1.13-fold increase in the *p*-PI3K/t-PI3K ratio, and a 0.71-fold decrease in the *p*-Akt/t-Akt ratio compared to the CON group. In the WSE group, in comparison to the DEX group, the *p*-mTOR/t-mTOR ratio exhibited a concentration-dependent increase of 1.02, 1.41, and 1.39-fold at concentrations of 25, 50, and 100 μg/mL, respectively. Additionally, the *p*-PI3K/t-PI3K ratio showed a significant increase of 1.41 and 1.39-fold at concentrations of 50 and 100 μg/mL, respectively. However, no significant difference was observed in the *p*-Akt/t-Akt ratio compared to the DEX group. Regarding SCE, utilized as a positive control, only the *p*-mTOR/t-mTOR ratio showed a significant difference from the DEX group at 1.05-fold (Figure 8A). Within the DEX group, there was no significant difference in MyoD protein expression compared to the CON, while Myogenin exhibited a noteworthy decrease. MyoD was increased by WSE treatment by 1.31, 1.35, and 1.39-fold, respectively, in a concentration-dependent manner, but there was no significant effect compared to the DEX group. Myogenin was increased by WSE treatment by 0.87, 0.97, and 0.96-fold, respectively, with a significant increase at 50 and 100 μg/mL compared to the DEX group (Figure 8B). FOXO3a, Myostatin, Atrogin-1, MuRF1 protein, and gene expression related to muscle degradation were examined (Figure 8C,D). DEX treatment significantly decreased the *p*-FoxO3a/t-FoxO3a ratio compared to the CON, while Atrogin-1 and MuRF1 showed significant increases at both protein and mRNA levels, and Myostatin showed a trend toward increased protein and a significant increase in mRNA. The *p*-FoxO3a/t-FoxO3a ratio exhibited a notable impact in the 100 μg/mL WSE (0.70-fold), showing a significant decrease compared to the DEX treatment group. For Myostatin, there was no significant change in protein by WSE but a significant decrease in mRNA (4.24, 3.90, and 2.61-fold, respectively) compared to DEX treatment. Atrogin-1 and MuRF1 proteins and mRNA showed concentration-dependent decreases. SCE, as a positive control, showed significant differences in Atrogin-1 and MuRF1 proteins and only Myostatin in mRNA. The protein expression of Sirt1 and PGC-1α exhibited an increase of 1.26-fold and 1.33-fold, respectively, in the DEX group compared to the CON group (Figure 8E). In the WSE group, the expression of Sirt1 increased in a concentration-dependent manner, measuring 1.36, 1.33, and 1.52-fold, respectively, with no significant change compared to the DEX group. Notably, the protein expression of PGC-1α showed a significant increase in the WSE 25, 50, and 100 μg/mL treatment groups by 2.07, 2.21, and 2.39-fold, respectively, compared to the DEX group. Remarkably, WSE at 50 and 100 μg/mL exhibited a more pronounced effect than SCE (2.01-fold).

## 4. Discussion

The etiology of muscle atrophy is multifaceted, and it manifests with age and aging even in the absence of specific muscle-related disorders (such as surgical injuries or metabolic conditions) [7]. This highlights the complex interplay between aging processes and the occurrence of muscle atrophy, extending beyond conditions directly impacting the muscles [8,25]. In the present study, our primary objective was to explore the influence of WSE on mechanisms associated with aging in skeletal muscle. We also aimed to confirm the dose-dependent effects by utilizing an aged mouse model that closely mimics sarcopenia, emphasizing a clinical perspective. Additionally, our investigation delved beyond factors directly linked to muscle physiology, encompassing aspects adversely affected by the aging process. This broader scope included the examination of chronic low-grade inflammation, muscle regeneration capacity, and mitochondrial biogenesis. As a positive control, *S. chinensis* extract (SCE) was utilized, with schizandrin as its main component. SCE has been demonstrated to be safe for both men and women over the age of 50. It has shown efficacy in enhancing muscle strength [26]. It has also shown beneficial effects in animal models of muscle atrophy under various conditions (e.g., dexamethasone and aging) [27,28]. Currently, it holds approval from the KFDA as a dietary supplement for the improvement of muscle strength.

The oral administration of WSE did not result in significant differences in body weight, diet, or water intake when compared to the CON. Moreover, in terms of toxicity parameters, there were no notable differences in ALT and AST levels following long-term administration between the control and treatment groups. Consequently, no toxicity associated with WSE was observed at the administered dosage.

Notably, WSE administration demonstrated a tendency to reduce both epidermal and abdominal fat, with a significant decrease observed at 200 mg/kg. Similarly, there was a tendency for an increase in muscle mass. The ratio of muscle to fat, adjusted for body weight, showed a tendency to increase across all treatment groups.

According to Hu et al. [29], serum levels of IL-1β, IL-6, and TNF-α in 12-week-old C57BL/6 mice were increased by 2.5, 2.8, and 4.5 times, respectively, compared with 8-week-old mice, and the results of our experiment also suggest that cytokine levels can be considered to increase with age. The administration of WSE led to a reduction in serum pro-inflammatory cytokines, specifically TNF-α and IL-1β. These alterations positively influence skeletal muscle, as inflammation can exacerbate muscle atrophy through the STAT and NF-κB pathways [30,31]. But the serum level of IL-6 was not significantly different. IL-6 signaling has been implicated in the stimulation of hypertrophic muscle growth and myogenesis by regulating the proliferative capacity of muscle stem cells. Furthermore, IL-6 demonstrates additional beneficial effects, including the regulation of energy metabolism, which is closely associated with the muscle’s capacity for active contraction and the synthesis and release of energy [9,32]. In our study, we assessed the same cytokines in both the gastrocnemius muscle and serum, revealing a significant reduction in IL-6 as well as IL-1β and TNF-α that were similarly reduced in serum by WSE. These findings suggest that WSE may have the potential to ameliorate age-dependent muscle inflammation.

Aging constitutes a significant risk factor for various musculoskeletal diseases, including osteoarthritis, osteoporosis, and sarcopenia, prompting widespread use of aging mouse models in sarcopenia research. The selection of 18-month-old mice provides a clinically relevant aging model [25]. Given the time-intensive process of establishing natural aging mouse models, an increasing number of researchers are turning to composite model methods to accelerate sarcopenia modeling. Many scholarly works have tackled this challenge by employing genetically induced aging models such as SAMP8 [33,34] or 12-month C57BL/6 mice [35,36,37], among others. The average lifespan of a mouse is 24 months, making a 12-month-old mouse roughly equivalent to a 50-year-old human. References employing 12-month-old mice as an experimental model also commonly refer to them as aged mice [38,39]. In constructing our experimental model, guided by preliminary experiments and references, we utilized 12-month-old mice, a stage marked by confirmed muscle loss.

Regarding the efficacy of WSE on skeletal muscle strength and mass, it effectively prevented the decline in muscle strength, resulting in a significant and stable maintenance of grip strength in the WSE 200 mg/kg groups. The weights of the three muscle tissues exhibited a tendency to increase, with a statistically significant difference observed in the WSE 200 mg/kg groups. However, at 300 mg/kg, certain markers were found to decrease. Apart from this study, a separate chronic oral toxicity study was conducted on WSE in a laboratory complying with good laboratory practice (GLP) approved by the Korean Food and Drug Administration (KFDA) and setting the no observed adverse effect level (NOAEL) at 5 g/kg. Therefore, it is argued that the effects observed at 300 mg/kg are not indicative of toxicity. Because natural products vary in composition, not all natural products exhibit dose-dependent effects, and some natural products exhibit an inverted U-shaped response in which efficacy tends to decrease at certain doses while increasing at higher doses [40]. WSE is also considered to demonstrate an inverted U-shaped response in the aging model, with the suggested optimal dose for the observed effect in this experiment being 200 mg/kg.

The aged mouse (CON) groups exhibited a decrease in running distance and time to exhaustion, both of which were subsequently restored in the WSE administration groups. These parameters were specifically associated with the slow-twitch muscle soleus, revealing a tendency for an increase in the WSE administration groups when compared to the CON groups.

As a mechanism study, three aspects of skeletal muscle aging were investigated: (1) the balance of muscle protein synthesis and degradation; (2) muscle fiber differentiation; and (3) mitochondrial biogenesis in skeletal muscle. The IGF-1/Akt/mTOR pathway determines muscle mass by regulating muscle protein synthesis and degradation. This pathway is activated by exercise but becomes less active with age, exacerbating muscle atrophy in the elderly [41,42]. The activity of the IGF-1/Akt/mTOR pathway was effectively increased by WSE, leading to the inhibition of muscle atrophy.

Dexamethasone-induced muscle atrophy results from elevated muscle protein degradation and reduced synthesis, as evidenced by the upregulation of myostatin promoter activity. This highlights the association between DEX-induced skeletal muscle atrophy and changes in myostatin regulation [11,43]. In our results, myostatin gene expression was significantly decreased by WSE treatment compared with the DEX group. Among the FOXO isoforms in skeletal muscle, FOXO3α is involved in the ubiquitin-proteasome system and the autophagy pathway [44,45]. Atrogin-1 and MuRF1 are crucial E3 ligase proteins that regulate the ubiquitination of proteins and their subsequent degradation in the proteasome within the cell [46,47].

Induction of myotube atrophy by DEX causes the dephosphorylation of Akt, resulting in the activation of FOXO and subsequent transcription of atrogin-1/MAFbx and MuRF1 [43,48]. A previous study reported that overexpression of MAFbx in myotubes caused atrophy, whereas mice deficient in either MAFbx or MuRF1 were resistant to atrophy [49]. Combined with our results, these findings illustrate that WSE may provide protection against DEX-induced myotube atrophy by enhancing the phosphorylation of AKT and FOXO3α. Furthermore, it diminishes muscle protein degradation through the phosphorylation of FOXO3a, inhibiting the transcription of MuRF1 and Atrogin-1.

The in vitro study revealed a promotive effect on myogenesis, particularly relevant to skeletal muscle, indicating an association with the PI3K/AKT signaling pathway, which plays a crucial role in proliferation and differentiation [50]. Hence, our focus was directed towards the PI3K/AKT pathway. PI3K (phosphatidyl inositol kinase) exists as a dimer, and upon binding to growth factor receptors like EGFR, it induces a structural transformation in AKT, activating the protein. Phosphorylated AKT, in turn, stimulates the mammalian targets of rapamycin (mTOR), facilitating protein synthesis. Numerous studies have demonstrated that heightened PI3K and AKT activity directly or indirectly promotes the induction of downstream myogenic proteins, such as Myogenin and MyoD, thereby expediting the differentiation and fusion of muscle cells [31,41,42].

According to our in vitro data, treatment with WSE resulted in a significant increase in phosphorylation levels of the PI3K/AKT/mTOR pathway. Additionally, WSE demonstrated an upregulation of myogenic regulatory proteins, namely MyoD and Myogenin, fostering an acceleration in the differentiation and fusion processes of myoblasts to form myofibers [51].

The SIRT1/PGC-1α signaling pathway is commonly known to align with the general pathway of mitochondrial biogenesis [52,53]. SIRT1, a key regulator of muscle metabolism and a potential therapeutic target for preventing muscle dysfunction, activates PGC-1α, a major controller of mitochondrial biogenesis. Elevated PGC-1α stimulates the upregulation of nuclear respiratory factors (NRF1 and 2) and mitochondrial transcription factor A (TFAM). NRF1 and 2 function as transcription factors that activate nuclear genes encoding various mitochondrial proteins, while TFAM governs mitochondrial DNA replication and transcription [54]. For this reason, many researchers are studying the SIRT1/PGC-1α signaling pathway as a factor related to mitochondrial biogenesis [55,56]. Therefore, for the maintenance of skeletal muscle mass, quality, and strength, the harmonious regulation of myogenic gene expression and signals in myoblasts is crucial. However, an imbalanced regulation of these factors can contribute to sarcopenia. Treatment with WSE was found to increase the levels of PGC1α and Sirt1, indicating that WSE upregulated mitochondrial biogenesis.

In our in vivo study, WSE administration increased muscle protein synthesis through the activation of the IGF-1/AKT/mTOR pathway and increased myogenic transcription factors such as Myogenin and MyoD. Also, WSE increased mitochondrial biosynthesis-related factors such as PGC1α and Sirt1.

However, the administration of WSE resulted in a marginal increase in MuRF1, Atrogin-1, and Myostatin expression. Withanone, a specific withanolide present in ashwagandha extract, induced a more robust differentiation of myoblasts into myotubes. It also facilitated the deaggregation of proteins that had formed aggregates due to heat and metal stress, concurrently activating hypoxia and autophagy pathways [57]. The differentiation of myoblasts necessitates functional degradative systems, including autophagy, which play a role in the formation of multinucleated terminally differentiated myotubes.

As a result, there was an increase in protein synthesis and mitochondrial biosynthesis. These findings indicate that WSE restored the imbalance between muscle protein synthesis and degradation induced by muscle atrophy, ultimately improving muscle mass. The functionality of skeletal muscle is intricately linked to both the content and function of mitochondria, factors that are governed by mitochondrial biogenesis [58].

This study aimed to investigate the potential ameliorative effects of WSE on a model of aging-induced muscular deterioration. The results unequivocally demonstrated the anti-sarcopenic impact of WSE in aging mice, shedding light on alterations in aging-related mechanisms. Consequently, pending the outcomes of clinical trials and further investigations, WSE could emerge as a viable candidate for preventing sarcopenia.

## 5. Conclusions

This study sought to explore the impact of WSE on age-related muscular deteriorations in aged mice. Aging induces chronic low-grade inflammation, disrupting the balance between muscle protein synthesis and degradation while diminishing the capacity for muscle regeneration and mitochondrial biogenesis. These changes contribute to muscle atrophy and worsen sarcopenia, yet WSE has demonstrated the potential to mitigate these physiological alterations. Specifically, WSE suppressed systemic low-grade inflammation by reducing serum IL-1β and TNF-α levels. Furthermore, muscle mass and strength witnessed an increase through the activation of the IGF-1/AKT/mTOR pathway. Additionally, WSE significantly elevated the expression of factors related to muscle regeneration and mitochondrial biogenesis, thereby contributing to the enhancement of muscle function, including exercise endurance.

## Figures and Tables

**Figure 1 nutrients-16-00157-f001:**
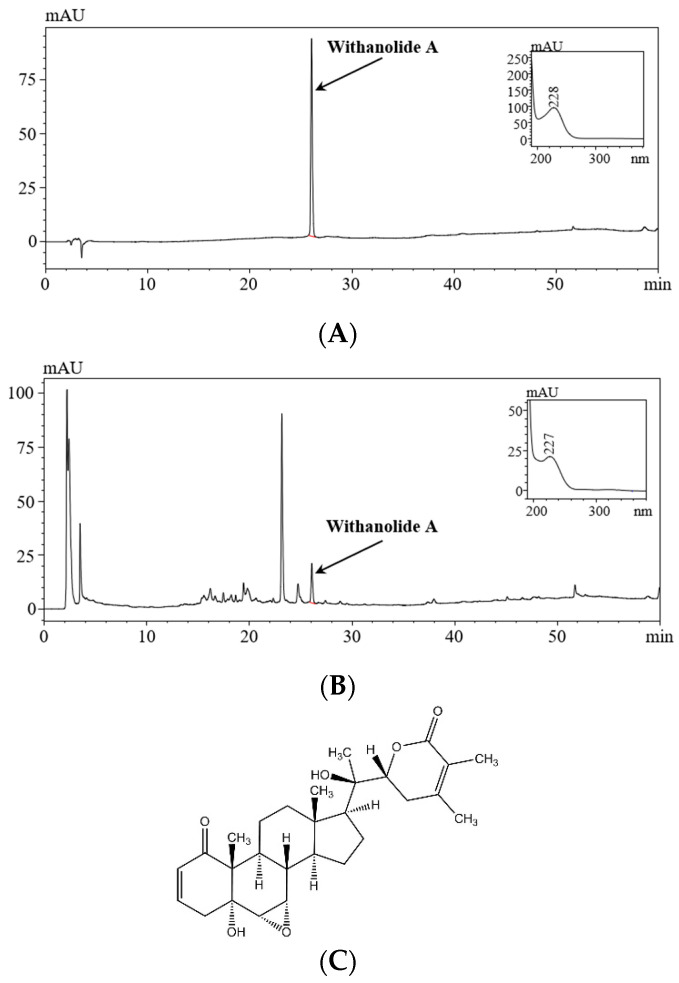
HPLC chromatogram of WSE and chemical structure of withanolide A. (**A**) withanolide A; (**B**) WSE; and (**C**) the structure of withanolide A.

**Figure 2 nutrients-16-00157-f002:**
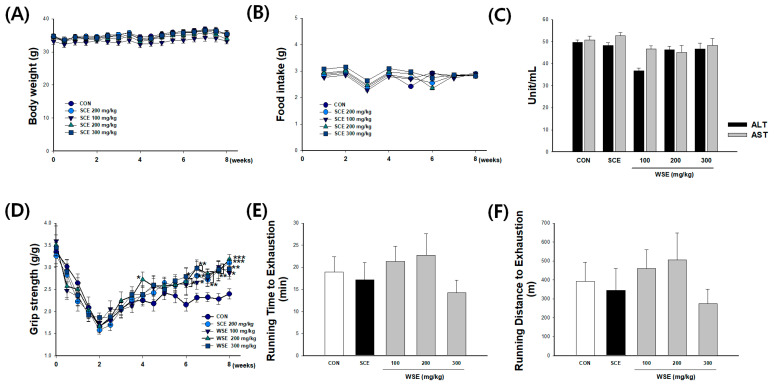
Effect of WSE on muscle performance in aged mice. (**A**) Body weight curves. (**B**) Food intake curves. (**C**) ALT and AST. (**D**) Grip strength curves. (**E**) Running time. (**F**) Distance to exhaustion time. The data are shown as mean ± SE; *n* = 9. * *p* < 0.05, ** *p* < 0.01, *** *p* < 0.001 versus the CON group.

**Figure 3 nutrients-16-00157-f003:**
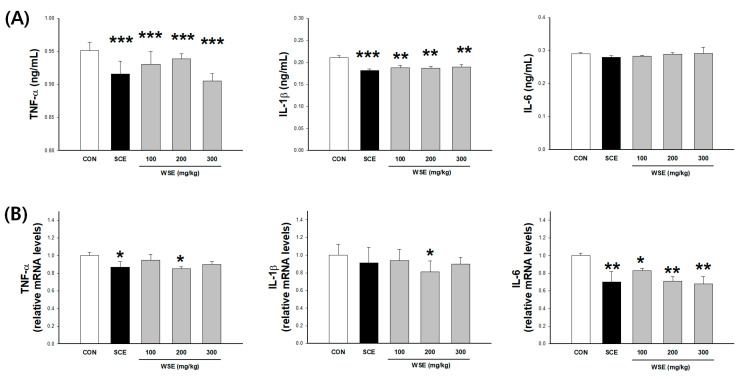
Effect of WSE on cytokine levels in aged mice. (**A**) Serum cytokine levels. (**B**) Expression levels of cytokine mRNA in the gastrocnemius. The data are shown as mean ± SE; *n* = 9. * *p* < 0.05, ** *p* < 0.01, *** *p* < 0.001 versus the CON group.

**Figure 4 nutrients-16-00157-f004:**
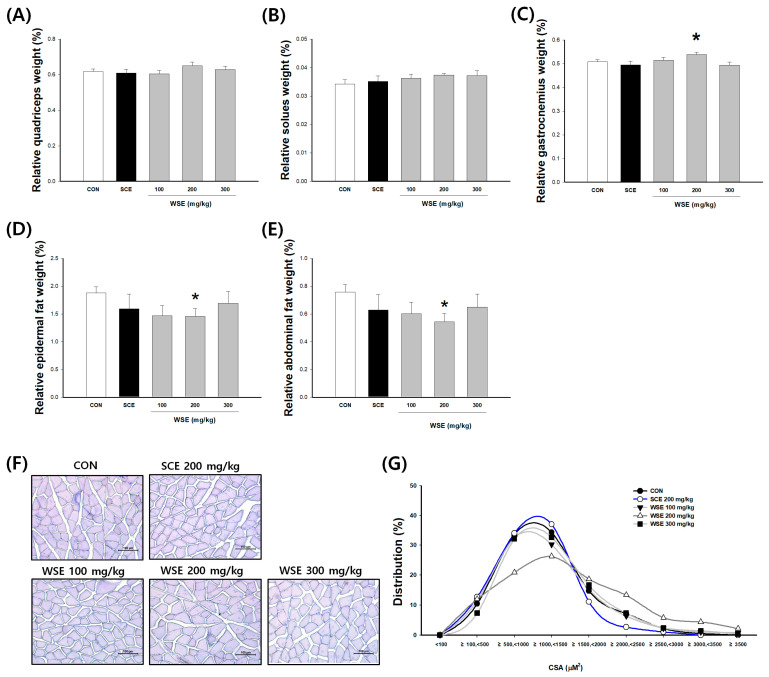
Effect of WSE on muscle, fat tissue mass, and myofiber cross-sectional area (CSA) in aged mice. The skeletal muscle mass of the (**A**) quadriceps, (**B**) soleus, and (**C**) gastrocnemius. The fat mass of (**D**) epidermal fat and (**E**) abdominal fat. It was shown as a ratio to the body weight (%). (**F**) Representative images of an H&E-stained gastrocnemius. (**G**) The distribution graph of muscle fiber CSA (%). The data are shown as mean ± SE; *n* = 9. * *p* < 0.05 versus the CON group.

**Figure 5 nutrients-16-00157-f005:**
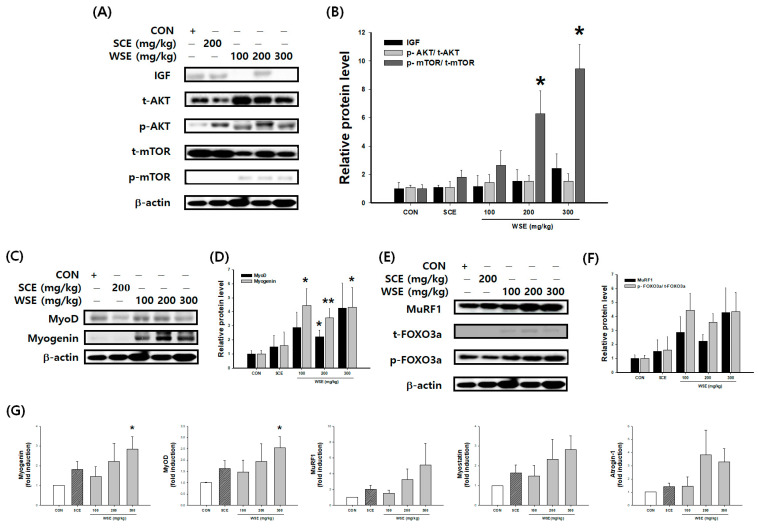
Effect of WSE on muscle protein synthesis and proteolysis through the AKT/mTOR pathway in aged mice. (**A**) Western blot images of IGF, AKT, and mTOR. (**B**) Relative protein expression of IGF, *p*-AKT/t-AKT, and *p*-mTOR/t-mTOR. (**C**) Western blot images of MyoD and Myogenin. (**D**) Relative protein expression of MyoD and Myogenin. (**E**) Western blot images of MuRF1 and FOXO3a. (**F**) Relative protein expression of MuRF1 and *p*-FOXO3a/t-FOXO3a. (**G**) Relative mRNA expression of Myogenin, MyoD, MuRF1, Myostatin, and Atrogin-1. The data are shown as mean ± SE; *n* = 6. * *p* < 0.05, ** *p* < 0.01 versus the CON group.

**Figure 6 nutrients-16-00157-f006:**
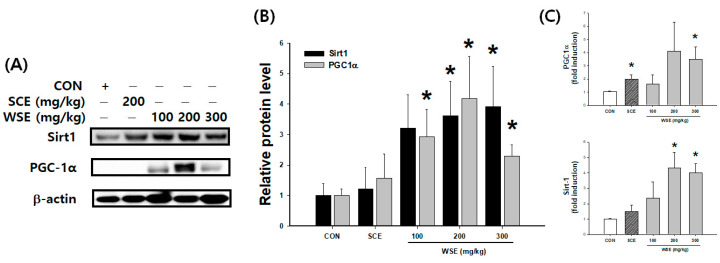
Effect of WSE on mitochondrial biogenesis through the Sirt1/PGC-1α pathway in aged mice. (**A**) Western blot images of Sirt1 and PGC-1α. (**B**) Relative protein expression of SIRT1 and PGC-1α. (**C**) Relative mRNA expression of Sirt1 and PGC-1α. The data are shown as mean ± SE; *n* = 6. * *p* < 0.05 versus the CON group.

**Figure 7 nutrients-16-00157-f007:**
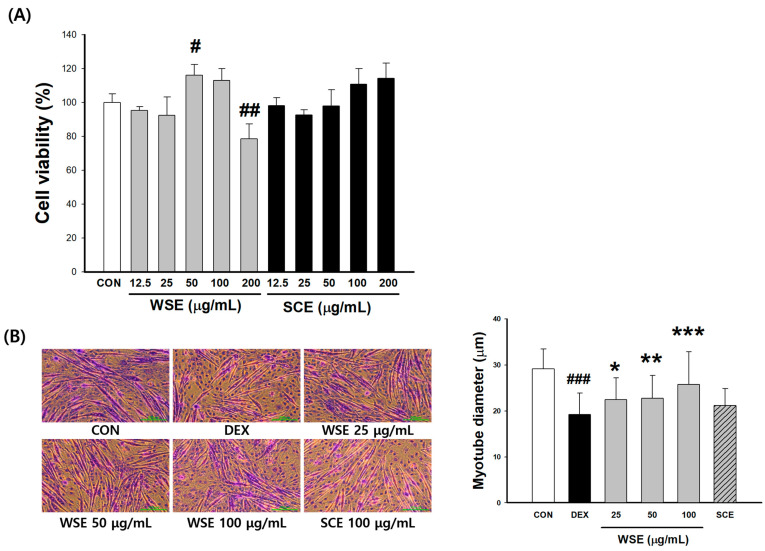
Effect of WSE on cell viability and myotube atrophy in C2C12 myotubes. (**A**) Cell viability following WSE treatment in C2C12 cells. (**B**) Changes in myotube diameter after WSE treatment of C2C12 cells with dexamethasone (DEX)-induced muscle atrophy. The data are shown as mean ± SE. # *p* < 0.05, ## *p* < 0.01, ### *p* < 0.001 versus the CON group, * *p* < 0.05, ** *p* < 0.01, *** *p* < 0.001 versus the DEX group.

**Figure 8 nutrients-16-00157-f008:**
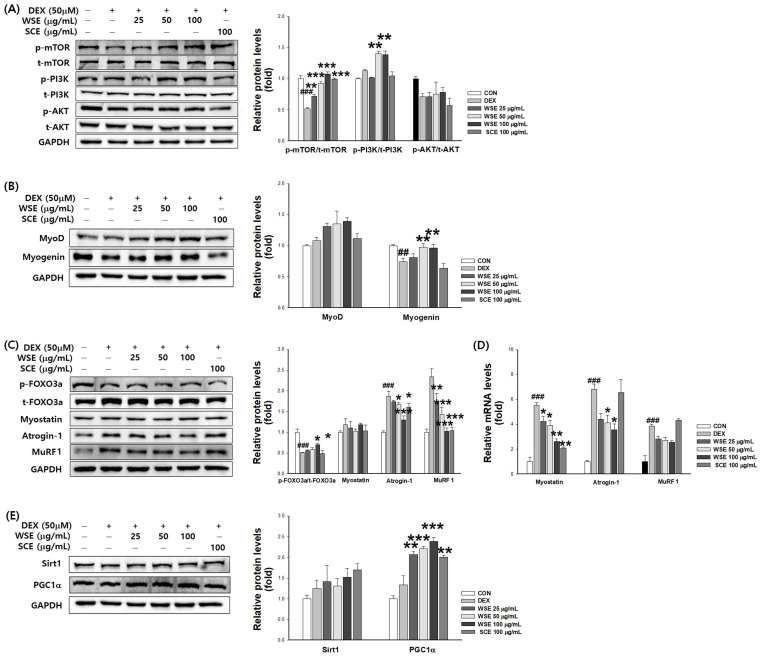
Effect of WSE on muscle protein synthesis and protein degradation through the PI3K/AKT pathway in dexamethasone-induced C2C12 muscle atrophy. The protein expression level of (**A**) mTOR, PI3K, and AKT, (**B**) MyoD and Myogenin, (**C**) FOXO3a, Myostatin, Atrogin-1, and MuRF1, (**E**) Sirt1 and PGC1α. (**D**) Relative mRNA level of Myostatin, Atrogin-1, and MuRF1. The data are shown as mean ± SE. ## *p* < 0.01, ### *p* < 0.001 versus the CON group, * *p* < 0.05, ** *p* < 0.01, *** *p* < 0.001 versus the DEX group.

**Table 1 nutrients-16-00157-t001:** The oligonucleotide primer sequences used in RT-qPCR.

Gene Name	Forward	Reverse
Myogenin	CAACTGCTCTGATGGCATGATGG	TGTTCTGCATCGCTTGAGGATGTC
MyoD	CAACTGCTCTGATGGCATGATGG	TGTTCTGCATCGCTTGAGGATGTC
MuRF1	AAGACTGAGCTGAGTAACTG	TAGAGGGTGTCAAACTTCTG
Atrogin-1	AGAAAGAAAGACATTCAGAACA	GCTCCTTCGTACTTCCTT
Myostatin	ACTGGACCTCTCGATAGAACACT	ACTTAGTGCTGTGTGTGTGGAGAT
Sirt1	CAAGATGCTGTTGCAAAGGAACC	CAAGATGCTGTTGCAAAGGAACC
PGC1α	AAGTGTGGAACTCTCTGGAACTG	GGGTTATCTTGGTTGGCTTTATG
TNF-α	CCCGAGTGACAAGCCTGTAG	GATGGCAGAGAGGAGGTTGAC
IL-1β	AGATGATAAGCCCACTCTACAG	ACATTCAGCACAGGACTCTC
IL-6	ACAGCCACTCACCTCTTCAG	CCATCTTTTTCAGCCATCTTT
β-actin	ATATCGCTGCGCTGGTCGTC	AGGATGGCGTGAGGGAGAGC

## Data Availability

The data presented in this study are available upon request from the corresponding author.

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
