# Peer review of "Ashwagandha Ethanol Extract Attenuates Sarcopenia-Related Muscle Atrophy in Aged Mice"

_nutrients, 2024, doi:10.3390/nu16010157_

Round 1

Reviewer 1 Report

Comments and Suggestions for Authors

The manuscript describes the anti-sarcopenia-related muscle atrophy effects of Withania somnifera extract (WSE) using aged mice. There is some useful information on sarcopenia prevention, while, several critical issues should be addressed.

1. So many mistakes in the References section, it is not acceptable.

2. WSE suppressed serum TNF-α, IL-1β in aged mice. Do aged mice have abnormal levels of inflammation related cytokines, or inhibition of the normal levels of the cytokines do not affect the related normal functions, evidences are essential.

3. Does author have batter images of a H&E-stained gastrocnemius, the quality of current one is quite bad (big white areas, water is not well controlled).

4. WSE enhances muscle protein synthesis and prevents protein degradation, author just analyzed the related signals, the detailed data on protein itself are needed (degradation can refer to ubiquitinylated protein assay).

5. Just the data on SIRT1/PGC-1α are not enough to conclude WSE has effect on mitochondrial biogenesis, detailed supportive data are important.

Comments on the Quality of English Language

English expressions can be more concise.

Author Response

Thank you very much for taking the time to review this manuscript. Please find the attached file for detailed responses.

Reviewer 2 Report

Comments and Suggestions for Authors

The article titled “Ashwagandha Ethanol Extract attenuates Sarcopenia-Related Muscle Atrophy in Aged Mice” describes the effects of Ashwagandha in skeletal muscles form C56BL/6 mice and in C2C12 cells.

The work is well organized and detailed, but there are some points that should be better developed.

1-    Authors should explain why They use S. chinensis extract (SCE) as positive control, references detailing its action in skeletal muscles should be added.

2-    One major concern relies on using C57BL/6 as a model that closely mimics sarcopenia, and therefore the conclusion that WSE prevents muscle aging. Actually a 48 weeks old mouse is comparable to a middle age human, and an 8 weeks treatment leads to 56 weeks, which is still located in the middle-aged time. Interestingly, there are recent works that analyze the fiber properties of tibialis anterior during aging (which is like gastrocnemius a fast muscle, containing a high percentage of MyHCIIB fibers, not found in humans), and show that they do not change dramatically from 8 to 18 months, while some modifications are more visible after 24 months of age. Therefore, data presented by the Authors suggest more an improvement of fiber size rather than the prevention of fiber size decrease. In conclusion, is WSE inducing hypertrophy? The direct effect of WSE on C2C12 myotubes further reinforces this concept.

Although, the results are well supported, C57BL/6 are not the best model for aging research. Maybe, modification of the title and correction of lines 402-404 could better frame the study.

3-    In the histological sections, especially WSE 300 mg/kg (FIG 4), it is possible to observe several small fibers. Is it possible that at this concentration WSE has a toxic effect?

In this context, in lines 427-429, Authors write that the dose identified was over 200 mg/kg. However, the dose 300 mg/kg seems to have not a positive effect in other parameters, such as IGF and PGC1-alpha expression. How can be this evidence explained?

4-    Authors report the decrease of inflammatory cytokines such as TNF-alpha and IL-1 beta, while there are not changes in IL-6, which is produced locally in the skeletal muscle. It cannot be excluded that reduction in TNF-alpha and IL-1 beta is not completely dependent on skeletal muscle. To undoubtedly test the reduction of inflammation, Authors should test the skeletal muscle tissue, by measuring its levels of inflammatory proteins.

5-    Minor points

a.      The keywords in vitro, in vivo are too general.

b.     Figure 5, some panels report actin and some others beta-actin, please correct or explain.

c.     In the graphs of figure 8, the bars are very small and the asterisks of the significance are not very readable.

Comments on the Quality of English Language

Minor editing required

Author Response

(The authors gave the same response as above.)

Round 2

Reviewer 1 Report

Comments and Suggestions for Authors

The comments raised by reviewers are almost addressed.

Comments on the Quality of English Language

No obvious errors, typos were found.